# Clinical Outcomes of the Endoscopic Step-Up Approach with or without Radiology-Guided Percutaneous Drainage for Symptomatic Walled-Off Pancreatic Necrosis

**DOI:** 10.3390/medicina59030569

**Published:** 2023-03-14

**Authors:** Tanawat Pattarapuntakul, Tummarong Charoenrit, Thanawin Wong, Nisa Netinatsunton, Bancha Ovartlarnporn, Thanapon Yaowmaneerat, Teeravut Tubtawee, Pattira Boonsri, Pimsiri Sripongpun

**Affiliations:** 1Gastroenterology and Hepatology Unit, Division of Internal Medicine, Faculty of Medicine, Prince of Songkla University, Hat Yai 90110, Songkhla, Thailandspimsiri@medicine.psu.ac.th (P.S.); 2Nantana-Kriangkrai Chotiwattanaphan (NKC) Institute of Gastroenterology and Hepatology, Faculty of Medicine, Prince of Songkla University, Hat Yai 90110, Songkhla, Thailand; 3Division of Radiology, Faculty of Medicine, Prince of Songkla University, Hat Yai 90110, Songkhla, Thailand

**Keywords:** directed endoscopic necrosectomy, endoscopic step-up approach, endoscopic transluminal drainage, percutaneous drainage, walled-off pancreatic necrosis

## Abstract

*Background and objectives:* Symptomatic walled-off pancreatic necrosis is a serious local complication of acute necrotising pancreatitis. The endoscopic step-up approach is the standard treatment for symptomatic walled-off pancreatic necrosis; however, adjunctive radiologic percutaneous drainage for this condition is controversial. This study compared the clinical and radiologic resolution of walled-off pancreatic necrosis achieved with the endoscopic step-up approach with or without radiology-guided percutaneous drainage. *Material and Methods:* This retrospective, single-centre cohort study enrolled patients with symptomatic walled-off pancreatic necrosis who underwent endoscopic transmural drainage (ETD) followed by directed endoscopic necrosectomy (DEN) with or without radiology-guided drainage. A total of 34 patients (endoscopic approach, *n* = 22; combined modality approach, *n* = 12) underwent the endoscopic step-up approach (ETD followed by DEN). Baseline characteristics, clinical success, and resolution of necrosis were compared between groups. *Results:* All patients achieved symptom resolution from walled-off pancreatic necrosis. The mean patient age was 58.4 years, and 21 (61.8%) were men. Following treatment with the endoscopic approach and combined modality approach, clinical success was achieved in 90.9% of patients within 11.5 days, and 66.7% of patients within 16.5 days, respectively. Both length of hospital stay (55 days vs. 71 days; *p* = 0.071) and time to complete radiologic resolution were shorter (93 days vs. 124 days; *p* = 0.23) in the endoscopic approach group. *Conclusion:* Both the endoscopic step-up approach and the CMD approach resulted in a favourably high clinical resolution rates in patients with symptomatic WON. However, clinical success rates seemed to be higher, and the length of hospital stay tended to be shorter in the endoscopic approach than in the CMD approach, as well as the significantly shorter necrosectomy time in each procedure was observed. Of note, these findings might be from some inherited differences in baseline characteristics of the patients between the two groups, and a randomized controlled trial with a larger sample size to verify these results is warranted.

## 1. Introduction

Walled-off pancreatic necrosis (WON) is a serious local complication of severe acute necrotising pancreatitis. This type of necrosis can be intra-pancreatic, peri-pancreatic, or both [1,2]. An acute necrotic collection may resolve gradually over time or progress to an encapsulated necrotic collection or WON, which typically occurs 4 weeks or more after the onset of acute pancreatitis [3]. WON can include various clinical presentations, such as abdominal pain, gastric outlet obstruction, jaundice, weight loss, and infection [4]. The treatment of symptomatic WON has undergone fundamental changes in recent years. Several studies have reported that the following minimally invasive approaches can achieve better outcomes: endoscopic transluminal drainage (ETD) with or without necrosectomy, laparoscopic or retroperitoneal surgical approach, and radiology-guided percutaneous approach followed by necrosectomy [5,6,7]. The endoscopic step-up approach, which consists of ETD followed by directed endoscopic transluminal necrosectomy (DEN), has been accepted as the standard treatment for symptomatic WON [5,8,9] as the clinical resolution rates of this approach are comparable to surgical necrosectomy and lower morbidity rates were observed [5,8]. The suitable duration of endoscopic treatment for WON is more than 4 weeks when completely encapsulated by a well-defined wall [10].

Treatment with DEN adjunct to ETD is associated with higher WON resolution rates (approximately 77% to 96%) and better safety than percutaneous drainage or ETD alone [11,12]. Percutaneous drainage in areas that are endoscopically inaccessible also results in improved clinical outcomes [7,13]. The PANTER trial demonstrated that the step-up approach involving percutaneous catheter drainage with subsequent minimally invasive surgical necrosectomy was superior to open surgical necrosectomy, which has complication rates ranging from 47% to 72% [9,14]. The percutaneous catheter drainage route was preferred over ETD for early necrosis (<4 weeks) caused by WON with incomplete wall encapsulation or endoscopically inaccessible areas [9]. Gluck et al. reported that combined modality drainage (ETD in combination with percutaneous drainage) is associated with a shorter length of hospitalisation and higher rates of complete resolution of WON than standard percutaneous drainage alone (96% vs. 80%) [15]. However, some retrospective data showed that the endoscopic step-up approach has a higher clinical success rate than combined modality drainage (86% and 58%, respectively) [13]. In this study, we compared WON resolution, including clinical success, clinical resolution, radiologic resolution, adverse events, number of necrosectomy sessions, achieved with the endoscopic step-up approach alone and the endoscopic step-up approach with radiology-guided percutaneous drainage.

## 2. Materials and Methods

This retrospective, single-centre cohort study enrolled all patients with symptomatic WON who underwent ETD followed by DEN with or without radiology-guided drainage in our centre, the only tertiary care university hospital in Southern Thailand, between January 2013 and June 2021.

The inclusion criteria were as follows: 1. Evidence of symptomatic completely encapsulated WON, e.g., abdominal pain, infection, sepsis, and inability to eat; 2. evidence of WON secondary to acute pancreatitis according to computed tomography (CT) or magnetic resonance imaging (MRI); 3. age of at least 18 years; 4. underwent ETD and DEN by EUS-guided drainage. Patients with uncorrectable coagulopathy were excluded. The disease severity and local complications were determined according to the CT severity index (CTSI) and the criteria of the revised Atlanta classification 2012 [1,2]. The consort diagram of the study was illustrated in Figure 1.

Generally, the routine practice for symptomatic WON treatment in our centre is the minimally invasive approach. The attending physicians (gastroenterologists, internists, and surgeons) referred symptomatic WON patients for ETD and DEN. All ETD and DEN procedures were performed by advanced endoscopists at the NKC institute of Gastroenterology and Hepatology in our centre. The adjunctive treatments with or without percutaneous drainage were justified according to clinical conditions, timing after pancreatitis, encapsulation of WON, and location of the collections by the attending physicians.

### 2.1. Study Definitions

Technical success was defined as the successful deployment of the Lumen-apposing metal stents (LAMSs) or at least one double pigtail plastic stent (DPPS) between the intestinal wall and WON. For the purposes of this study, only WON patients who achieved successful deployment followed with DEN were included in the analysis. Clinical resolution was defined as improvement in the sign and symptoms of systemic inflammatory response syndrome (SIRS), sepsis, abdominal pain, and able to resume oral diet after the intervention. Clinical success was considered as successful ETD and DEN, defined as a decrease in the size of WON to <3 cm on cross-sectional abdominal imaging, with resolution of symptoms within 6 months of follow-up [16].

### 2.2. Endoscopic Transmural Drainage

EUS was performed using a linear array echoendoscope (GF-UCT/P-180 series; Olympus Medical System, Corp., Tokyo, Japan) and ultrasound machine (model SSD alpha 10; Aloka, Tokyo, Japan). Prior to the procedure, the indication for drainage and cross-sectional abdominal imaging were reviewed and WON was identified by EUS. The decision to perform EUS-guided drainage regarding the puncture site, needle size, and stent type were determined at the discretion of the endoscopists. The optimal location of ETD via the transgastric or transduodenal approaches was chosen under EUS and Doppler guidance to ensure a minimal distance between WON and the intestinal wall and to avoid blood vessels. The puncture was performed with a 19-gauge needle (Echotip; COOK Endoscopy, Winston-Salem, NC, USA). Once the proper position of the tip of the needle in WON was identified, the stylet was removed. Thereafter, the collection was aspirated, and the fluid was subjected to bacterial gram staining and culture testing. The 0.025-inch guidewire was subsequently coiled into the collection under EUS guidance, and the access site was dilated using a cautery method with a 6-Fr cystotome followed by a 6-mm hurricane balloon dilator using the noncautery method. One or two 7-Fr double pigtail stents with a length of 5 cm were inserted into WON or LAMS (stent size 10 × 30 mm; NAGI; Teawong, Korea) and placed between the gastroduodenal lumen and the collection, as shown in Figure 2.

### 2.3. Percutaneous Drainage

In patients who underwent adjunctive percutaneous drainage, the procedure was carried out by interventional radiologists using ultrasound or CT guidance. The drainage catheters were positioned within the necrotic fluid collections while attempting to avoid pulmonary, hepatic, colonic, and vascular structures. Thereafter, the aspirated fluid was subjected to bacterial gram staining and culture testing, and a 12-Fr to 15-Fr catheter was placed into the collection to perform drainage. The aspiration was attempted to obtain as much fluid as possible, the drainage catheters were then subjected to gravity and irrigated with 10 to 20 mL of sterile saline three times daily. Percutaneous catheters were sequentially up-sized to a maximum of 18 Fr.

### 2.4. Directed Endoscopic Transmural Necrosectomy

Endoscopic necrosectomy aims to remove the tissue debris and infected material and multiple open dead spaces that contain infected material. The procedure was performed under conscious sedation by experienced endoscopists using a gastroscope (EVIS EXERA III, GIF-1TH190; Olympus Medical System, Corp., Tokyo, Japan).

The necrosectomy was performed after the ETD. The technique of necrosectomy includes mechanical removal using a snare, basket, or tripod retriever and intermittent saline irrigation, followed by 200 mL diluted hydrogen peroxide (1:1) at the end of procedure. DEN was performed repeatedly until pink granulation tissue was demonstrated in the wall of the collection, as shown in Figure 2.

At our centre, endoscopic retrograde cholangiopancreatography with pancreatic duct stent placement is not routinely performed. This procedure is performed only in the setting of a pancreatic fistula, unresolved or delayed collection over time, pancreatic stricture, and evidence of disconnected duct syndrome. Additionally, the multiple transmural gateway technique (MTGT) approach is not a routine practice because placement of the LAMS has become the first-line deployment at our centre.

### 2.5. Data Collection

Using the hospital’s electronic database, we collected the following demographic and clinical characteristics: age, sex, cause of pancreatitis, initial laboratory data, cross-sectional abdominal imaging data, disease severity, and local complications according to the CTSI and revised Atlanta classification 2012, EUS procedure data, radiology-guided drainage procedure data, stent types, clinical and radiologic resolution, and length of hospital stay.

### 2.6. Statistical Analysis

The patients were categorized into the endoscopic approach group (ETD with DEN only) or the combined modality drainage group (CMD; ETD with DEN plus percutaneous drainage). Baseline characteristics (demographic, clinical, and laboratory data) and the outcomes between the two groups were compared using the Wilcoxon test for non-normally distributed data and Student’s *t*-test for normally distributed data. The categorical data were compared using the chi-square test or Fisher’s exact test. *p* < 0.05 was considered statistically significant. All statistical analyses were performed using R program version 4.1.0 (R Foundation for Statistical Computing, Vienna, Austria).

## 3. Results

During the study period, there were 60 cases of symptomatic WON in our centre. Of those, we included 34 eligible patients (21 males; mean age, 58.4 ± 12 years) (Figure 1), there were 22 patients in the endoscopic approach group and 12 patients in the CMD group. Table 1 shows the baseline characteristics of the patients in the entire cohort as well as the characteristics when categorized into the two groups. For the entire cohort, the most common aetiology of pancreatitis was gallstones (56%), followed by alcohol (29%). Age, sex, body mass index, aetiology of pancreatitis, disease severity, comorbid disease, and baseline laboratory test results were not significantly different between the two groups. A higher proportion of patients with multiorgan dysfunction were observed in the CMD group than in the endoscopic approach group, but not statistically different.

The detail regarding WON of the patients in the cohort are shown in Table 2. The severity of acute pancreatitis according to the CTSI and revised Atlanta classification was high in both groups, the median CTSI was 10 in the endoscopic approach group, whereas it was 9 in the CMD group (*p* = 0.039). WON was well encapsulated approximately 30 days after onset and mostly located centrally and near the stomach, and the mean WON size was comparable in both groups. The most frequent symptoms of WON were infection, abdominal pain, and gastric outlet obstruction, in decreasing order, respectively. The collection at the left paracolic gutter was observed in the CMD group more than in the endoscopic approach group, yet it was not statistically significant. Vascular thrombosis, including portal vein, splenic vein, and superior mesenteric vein thrombosis was found quite frequently in this study.

Endoscopic drainage was initially performed after the onset of pancreatitis at the mean of 38 days in the endoscopic approach and at a mean of 42.5 days in the CMD group. The average time to necrosectomy after drainage was around 8 days in both groups. The detail about the endoscopic procedure and adverse events between the two groups are shown in Table 3. All patients underwent a transgastric approach for endoscopic drainage. A LAMS was used for the drainage in approximately 80% of this cohort. Surprisingly, the mean number of necrosectomy procedures was equal in both groups (average, 3.5 times in each group). This procedure is usually performed at our centre with additional hydrogen peroxide for chemical debridement, and it accounts for 60% of combined mechanical debridement procedures. The mean total necrosectomy time was higher in the CMD group (approximately 118 min) than in the endoscopic approach group (78 min; *p* < 0.001). Additionally, minor complications of ETD and DEN occurred equally in both groups, such as bleeding or perforation; they were treated with endoscopic and conservative treatment.

All instances of symptomatic WON achieved clinical resolution after the intervention. Clinical success, radiologic resolution, and length of hospital stays between the endoscopic approach group and the CMD group are shown in Figure 3. Following the intervention, clinical success and symptoms resolution was achieved in 90.9% of patients within 11.5 days in the endoscopic approach group, tended to be better than that in the CMD group in which clinical success was observed in 66.7% of patients within 16.5 days. Furthermore, the time to complete radiologic resolution was shorter in the endoscopic approach group (93 days) than in the combined modality drainage group (124 days). Additionally, the total length of the hospital stay tended to be shorter in the endoscopic approach group than in the CMD group (mean 54.6 vs. 70.6 days, *p* = 0.071). However, all of the aforementioned differences in outcomes were not statistically different.

## 4. Discussion

Infected WON is a life-threatening condition. The main treatment to improve overall survival is optimal drainage with or without necrosectomy. Endoscopic drainage is less invasive than surgical necrosectomy and is the current standard minimally invasive endoscopic modality [17]. The endoscopic step-up approach has been designed so that ETD followed by DEN can be performed if needed, allowing for clinical success rates that range between 75–90% [18]. This procedure can achieve complete clinical and radiologic resolution with lower mortality rates in comparison with open necrosectomy (risk ratio 0.27; 95% CI 0.08–0.88; *p* = 0.03) [19].

In this study, we evaluated the clinical outcomes of the endoscopic approach and compared them with the outcomes of the CMD approach for symptomatic WON. The baseline characteristics of the endoscopic approach and CMD approach groups were similar except that the median CTSI in the endoscopic approach group was higher than in the CMD group (10 vs. 9, *p* = 0.039). Interestingly, WON patients in our study seemed to have a greater severity grade according to the CTSI (median, 9–10) compared to other prior studies (median, 7–8) [13,15,20]. The mean size of WON was 14 cm and located centrally, which is slightly larger than that reported previously as well [13,21].

We did not include WON patients who underwent percutaneous drainage alone in this study, as previous studies showed that clinically successful percutaneous drainage for symptomatic WON was achieved in only 35% to 51% of cases [8,14], which were inferior to the endoscopic step-up approach [15,22]. For the ETD procedure, a LAMS (NAGI stent) was used in approximately 80% of our cohort. A previous study showed that LAM stents were superior to plastic stents in terms of overall treatment efficacy and number of endoscopy sessions (2.2 vs. 3.6; *p* = 0.04) [16,21,23]. The larger lumen diameter stents allow for adequate drainage and prevent occlusion and subsequent infection, which were strengths of LAM over DPPS.

The endoscopic necrosectomy procedure was performed an average of 3.5 times in both groups; however, the total necrosectomy procedure time of each session was significantly higher in the CMD group (118 min) than in the endoscopic approach group (78 min). These findings may be partly explained by the location of the collection in which the collection at the right and left paracolic gutters were more prevalent in the CMD group than in the endoscopic approach group, albeit not statistically significant. It might lead to the deeper penetrating route of the collection in the CMD group and make it more time-consuming.

All patients in our study achieved clinical resolution after the procedures. The endoscopic approach resulted in earlier clinical resolution, within 11 days, compared to the CMD group in 16 days. Interestingly, the endoscopic approach tended to achieve higher clinical success rates of 90.9% than that in the CMD approach (66.7%) despite a higher baseline CTSI. The clinical success rate in the endoscopic approach group in our study was akin to the study by Siddiqui et al. in which ETD followed by DEN with LAM demonstrated a high endoscopic therapy success of up to 88.2% [24]. Furthermore, our study showed the trend of a shorter hospital stay in the endoscopic approach group than in the CMD group (54.6 vs. 70.6 days, *p* = 0.071), this finding is in accordance with that reported by Nemato et al., showing that the endoscopic approach was associated with a reduced hospital stay of approximately 17 days, and that the dual modality approach was associated with a hospital stay of approximately 31 days [13]. The satisfactory high clinical success rates and favourable outcomes of the patients in the endoscopic approach group in our study might be attributed to a couple reasons; first, the greater tendency of the centrally located WON (non-complex WON) in the endoscopic approach group, which might be easier to drain and DEN than cases of complex WON located in areas that are inaccessible for an endoscopic approach; in addition, the large lumen patency of the stent (LAMS) in our cohort might be a beneficial effect for adequate drainage and clinical outcomes; and lastly, the H_2_O_2_-assisted DEN in approximately 80% of our cases may also play a role, the previous meta-analysis showed that H_2_O_2_-assisted DEN achieved a high clinical success of 91.6% (95% CI 86.1–95) and no adverse events attributable to H_2_O_2_ were reported [25].

Although patients in the CMD group had less severe disease than the endoscopic approach group according to the initial CTSI, the higher number of collections in the right and left paracolic gutters may contribute to the numerically lower clinical success rate in the CMD group rather than the endoscopic approach group. Generally, radiologic drainage is indicated for cases of early sepsis that do not respond to medication and cases of gas formation in the collection, and additional drainage is indicated when the area is endoscopically inaccessible. The major concern of percutaneous drainage was external pancreatic fistula (EPF), Rana SS, et al. showed that the incidence of EPF was significantly higher in the percutaneous drainage (21.95% vs. 0%, *p* = 0.021) compared with ETD [26]. In our cohort, percutaneous necrosectomy was necessary for only one patient; in that patient, a 28-Fr catheter was placed via the intercostal chest tube to perform drainage, followed by an 8.8-mm-diameter gastroscope for mechanical necrosectomy. There was no EPF observed in our cohort.

Complications, such as perforation and bleeding were not different between the two groups. Stent-related complications, including delayed bleeding and buried LAMS syndrome was not observed in this study; however, the stent indwelling time (73 days) in our study was longer than that observed in previous studies [27]. Fortunately, no disease-related death occurred in this study.

### Limitations

This study represents a real-world situation of symptomatic WON patients in developing countries, where patients usually present late during the course of disease with abdominal pain and a large collection demonstrated by the high CTSI and large size WON in both groups. Although the clinical resolution and clinical success rates were satisfactory in our study, we acknowledge some limitations of the current study. The study is retrospective, non-randomized in nature, therefore, some inherited differences in baseline characteristics resulting in the selection bias of the patients to receive either treatment approach inevitably existed. Although a statistically significant level was not reached, the number of patients with the extension of collection to the left and right paracolic gutters were noticeably higher in the CMD group and this might have a substantial impact on the outcomes, especially for the difference in the necrosectomy time between the two groups, as shown in the study. The sample size of patients with WON who underwent endoscopic treatment in our study is rather small. The differences in both baseline characteristics and outcomes may be statistically significant if the sample size becomes larger. Nonetheless, we think that our study is useful in terms of it as a reflection of a real-world practice in a country with limited resources. The protocol for the procedure (ETD and DEN), follow-up, clinical condition, and imaging after clinical resolution were consistent. Therefore, the data and follow-up were accurate and complete in the present study.

## 5. Conclusions

Both the endoscopic step-up approach and the CMD approach resulted in a favourably high clinical resolution rates in patients with symptomatic WON. However, clinical success rates seemed to be higher, and the length of hospital stay tended to be shorter in the endoscopic approach than in the CMD approach, and a significantly shorter necrosectomy time in each procedure was also observed. Of note, these findings might be from some inherited differences in baseline characteristics of the patients between the two groups, and a randomized controlled trial with a larger sample size to verify these results is warranted.

## Figures and Tables

**Figure 1 medicina-59-00569-f001:**
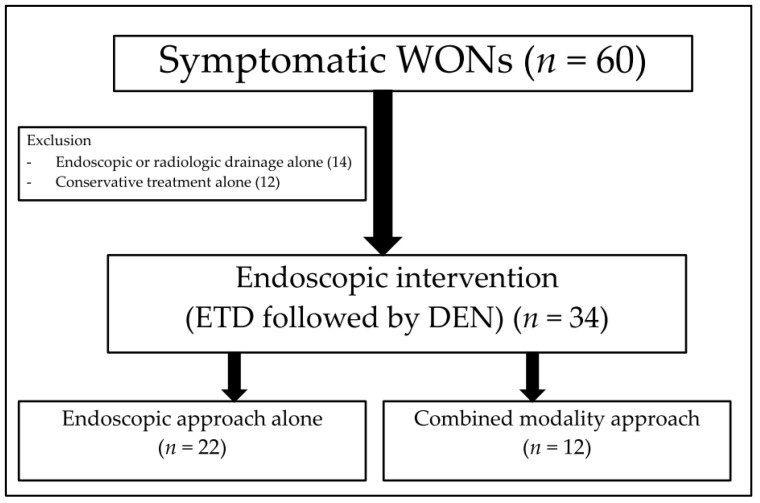
The consort diagram of the study.

**Figure 2 medicina-59-00569-f002:**
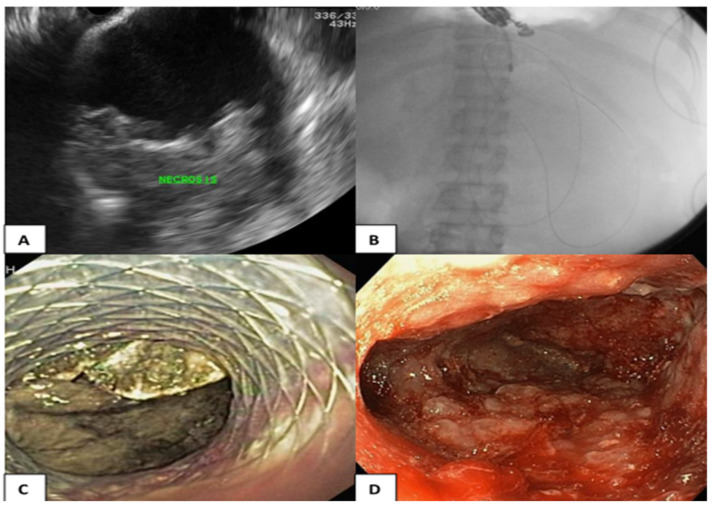
(**A**) Endosonographic view of the WON; (**B**) The A guidewire and a cystotome catheter (cautery method) advancing into the WON; (**C**) LAMS was placed between the gastric lumen and the collection for ETD; (**D**) DEN was performed repeatedly until pink granulation tissue was demonstrated in the wall of the collection.

**Figure 3 medicina-59-00569-f003:**
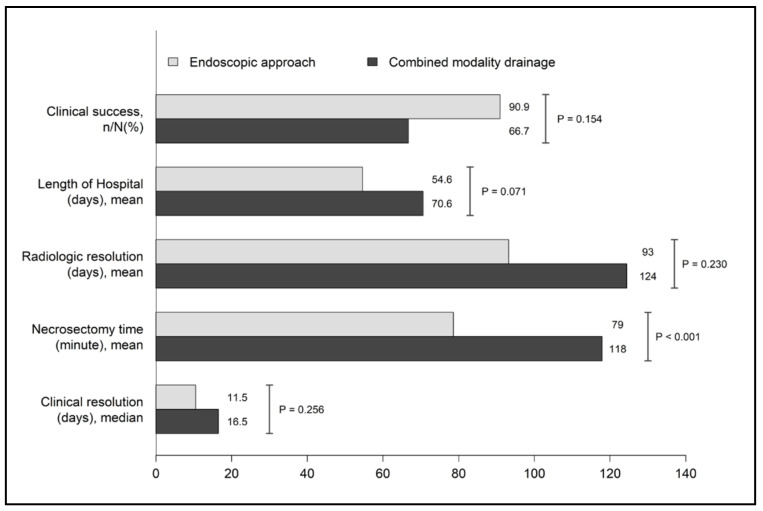
Clinical outcomes between the endoscopic approach and combined modality drainage.

**Table 1 medicina-59-00569-t001:** Baseline characteristics of the patients in the study.

Variables	All Cohort (*n* = 34)	Endoscopic Approach (*n* = 22)	Combined Modality Drainage (*n* = 12)	*p* Value
Sex (male), *n* (%)	21 (61.8)	14 (63.6)	7 (58.3)	1
Age (year) *	58.4 ± 12	60.6 ± 12.6	54.4 ± 10.1	0.156
BMI (kg/m^2^) *	23.4 ± 4.2	24.1 ± 4.2	22.1 ± 4	0.199
Aetiology of pancreatitis, *n* (%)				0.79
Gallstones	19 (55.9)	13 (59.1)	6 (50)	
Alcohol	10 (29.4)	6 (27.3)	4 (33.3)	
Post-ERCP	4 (11.8)	2 (9.1)	2 (16.7)	
Others	1 (2.9)	1(4.5)	0	
Severity of pancreatitis *				0.791
(Revised-Atlanta criteria), *n* (%)				
Moderately severe	9 (26.5)	7 (31.8)	2 (16.7)	
Severe	25 (73.5)	15 (68.2)	10 (83.3)	
Multi-organ dysfunction, *n* (%)	16 (47)	9 (40.9)	7 (58.3)	0.54
Comorbid disease, *n* (%)	23 (67.6)	15 (68.2)	8 (66.7)	1
Hypertension	19 (55.9)	13 (59.1)	6 (50)	0.882
Diabetes mellitus	12 (35.3)	8 (36.4)	4 (33.3)	1
Dyslipidemia	13 (38.2)	9 (40.9)	4 (33.3)	0.727
Ischemic heart disease	1 (2.9)	1 (4.5)	0	1
Cerebrovascular disease	1 (2.9)	1 (4.5)	0	1
others	3 (8.7)	2 (9)	1 (8.3)	1
Antiplatelet use, *n* (%)	4 (11.8)	3 (13.6)	1 (8.3)	1
Initial eGFR, *n* (%)				0.138
>60	25 (73.5)	18 (81.8)	7 (58.3)	
30–60	7 (20.6)	4 (18.20)	3 (25)	
<30	2 (5.9)	0	2 (16.7)	
Initial total bilirubin † (mg/dL)	0.8 (0.6–1.5)	0.8 (0.6–1.4)	1.1 (0.6–1.8)	0.601
Initial albumin * (mg%)	3.5 ± 0.7	3.6 ± 0.6	3.4 ± 0.8	0.594
Initial platelet count (×10^3^) *	312 ± 98	311 ± 100	314 ± 103	0.925
Initial hematocrit (%) *	36.5 ± 8.7	36.9 ± 7.6	35.9 ± 10.9	0.766
Initial amylase † (mg/dL)	1176 (782- 2545)	1241 (660–2598)	1089 (862–2105)	0.514
Initial lipase † (mg/dL)	2739 (126–5730)	3787(116–7417)	1451 (157–3692)	0.514
Positive hemoculture, *n* (%)	2 (5.9)	1 (4.5)	1 (8.3)	0.601

* Data are expressed as the mean ± SD. † Data are expressed as median (IQR). BMI, body mass index; eGFR, estimated glomerular filtration rate; ERCP, endoscopic retrograde cholangiopancreatography; IQR, interquartile range; SD, standard deviation. The *p*-values shown in the table are the comparisons between the endoscopic approach and the CMD group.

**Table 2 medicina-59-00569-t002:** Clinical and radiological characteristics of walled-off pancreatic necrosis.

Variables	All Cohort (*n* = 34)	Endoscopic Approach (*n* = 22)	Combined Modality Drainage (*n* = 12)	*p* Value
Severity of pancreatitis by CTSI †	10 (6.5–10)	10 (8.5–10)	9 (6–10)	0.039
Initial local complication (72 h)				
APFC	2 (5.9)	2 (9.1)	0 (0)	0.529
ANC	31 (91.2)	19 (86.4)	12 (100)	0.537
Encapsulated WON after diagnosis of pancreatitis, (day) †	30.5 (20–43.8)	28.5 (20–45)	32 (22–36)	1
Symptoms of WON, *n* (%)				
Infected WON	32 (94.1)	20 (90.9)	12 (100)	0.529
Abdominal pain	17 (50)	9 (40.9)	8 (66.7)	0.282
Gastric outlet obstruction	2 (5.9)	1 (4.5)	1 (8.3)	1
Intolerable to eat	1 (2.9)	1 (4.5)	0	1
Size of WON (CT scan/MRI), (cm) *	14.6 ± 3.7	14.7 ± 3.6	14.5 ± 4	0.909
Other CT findings of WON, *n* (%)				
Vascular thrombosis	31 (91.2)	20 (90.9)	11 (91.7)	1
Pseudo-aneurysm in WON	2 (5.9)	1 (4.5)	1 (8.3)	1
Completely walled-off	21 (61.8)	12 (54.5)	9 (75)	0.292
Wall thickening (mm) †	3 (3–4)	4 (3–4)	3 (3–4)	0.493
Location, *n* (%)				
stomach	34 (100)	22 (100)	12 (100)	0.086
Rt paracolic gutter collection	6 (17.6)	3 (13.6)	3 (25)	0.641
Lt paracolic gutter collection	12 (35.3)	5 (22.7)	7 (58.3)	0.062
Presence of air in WONs	15 (44.1)	11 (50)	4 (33.3)	0.566

* Data are expressed as the mean ± SD. † Data are expressed as median (IQR). ANC, acute necrotic collection; APFC, acute peripancreatic fluid collection; CT, computed tomography; CTSI, computed tomography severity index; IQR, interquartile range; Lt, left; MRI, magnetic resonance imaging; Rt, right; SD, standard deviation; WON, walled-off pancreatic necrosis. The *p*-values shown in the table are the comparisons between the endoscopic approach and the CMD group.

**Table 3 medicina-59-00569-t003:** Procedure techniques and adverse events.

Variables	All Cohort (*n* = 34)	Endoscopic Approach (*n* = 22)	Combined Modality Drainage (*n* = 12)	*p* Value
Duration of first drainage after the diagnosis of AP (day) †	38 (24.5–71)	38 (24.5–61)	42.5 (32–87.2)	0.773
Duration of first necrosectomy after drainage (day) †	8 (5–12.5)	7.5 (5.2–11)	8 (4.5–14)	0.828
- Location of drainage, *n* (%)				0.433
- Lesser curvature of stomach	23 (67.7)	13 (59)	10 (83.4)	
- Greater curvature of stomach	7 (20.6)	5 (22.7)	2 (16.7)	
- Antrum of the stomach	4 (11.8)	4 (18.2)	0	
Stent type, *n* (%)				0.317
Single plastic stent	2 (5.9)	1 (4.5)	1 (8.1)	
Multiple plastic stents	5 (14.7)	2 (9.1)	3 (25)	
LAM stents	27 (79.4)	19 (86.4)	8 (66.7)	
MTGT, *n* (%)	2 (5.9)	1 (4.5)	1 (8.3)	1
- Complications of ETD, *n* (%)				1
- Bleeding, *n* (%)	8 (23.5)	5 (22.7)	3 (25)	
Total necrosectomy time, min (SD) *	92.5 ± 34	78.6 ± 27.4	118 ± 31	<0.001
Number of DEN, *n* (IQR) †	3.5 (2–5)	3.5 (2–5)	3.5 (2.8–5)	0.839
Necrosectomy technique, *n* (%)				
- Snare alone	7 (20.6)	4 (18.2)	3 (25)	0.677
- Snare with chemical irrigation (H_2_O_2_)	21 (61.8)	13 (59.1)	8 (66.7)	0.727
Complications of DEN, *n* (%)				
Bleeding	7 (20.6)	5 (22.7)	2 (16.7)	1
Perforation	1 (2.9)	0	1 (8.3)	0.353

* Data are expressed as the mean ± SD. † Data are expressed as median (IQR). AP, acute pancreatitis; DEN, directed endoscopic transluminal necrosectomy; ERP, endoscopic retrograde pancreatography; IQR, interquartile range; LAM, lumen-apposing metal; MTGT, multiple transluminal gateway technique; P duct: pancreatic duct; SD, standard deviation. The *p*-values shown in the table are the comparisons between the endoscopic approach and the CMD group.

## Data Availability

Data and materials will be provided upon reasonable request due to privacy and ethical restrictions policy.

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
