# Peer review of "Clinical Outcomes of the Endoscopic Step-Up Approach with or without Radiology-Guided Percutaneous Drainage for Symptomatic Walled-Off Pancreatic Necrosis"

_medicina, 2023, doi:10.3390/medicina59030569_

Round 1

Reviewer 1 Report

1.       The topic can definitely raise the interest among surgeons, gastroenterologists and interventional radiologists

2.       The major problem is the patient selection bias - the groups of patients, who received two different type treatments are not identical.  It’s partially mentioned in the “Limitations” section. So, the treatment effectiveness is conditioned mainly not by treatment type, but by pre-treatment patient status. Therefore, conclusion is not relevant; paper needs to be rewrited.

3.       Imaging should be used to demonstrate performed interventions

Author Response

Thank you for your comments. 

We have explained and adjusted the crucial reports following the comments as the attached file 

We appreciate in your valuable comments and I look forward to receiving your respond.

Best regards.

Tanawat Pattarapuntakul MD.

Reviewer 2 Report

Revision

This study is low powered due to the small number of cases

This is not a comparative study as both groups are no comparable and the advantages of the pure endoscopic group are attributed to the more favourable conditions of their cases (centrally located, no extension to paracolic gutters, accessible) in comparison to radiology guided group.

Introduction: Adjunctive with DEN is associated with higher WON resolution rates (approximately 77% to 96%) and better safety than percutaneous drainage or ETD alone [13,14]

Aim

In the aim, all patients should be listed as one group and I prefer removing the comparison part, and mentioning the radiology guided drainage as a subset of patients.

methods:

the authors mentioned transgastric or transduodenal routes. Then, all patients were managed transgastrically.

Discussion

Endoscopic and radiologic drainage were (are) less invasive than surgical necrosectomy and are the current standard minimally invasive endoscopic modalities [21]. In these cases

Both groups are not the same.

this treatment comprised early and adjunctive (delayed) percutaneous drainage

I couldn’t understand this sentence: The majority concern 312 percutaneous drainage was external pancreatic fistula (EPF) a 28-Fr catheter was placed via the intercostal chest

General:

·         The manuscript can be shortened with removal of repetitions

·         Grammar and language check

Conclusion: The endoscopic step-up approach resulted in the clinical resolution of symptomatic walled-off pancreatic necrosis comparable to that of the combined modality drainage. However, the endoscopic approach alone allows higher clinical success, early clinical and radiologic resolution, and a shorter hospital stay.

You should specify subset of patients who were treated by percutaneous drainage

Author Response

(The authors gave the same response as above.)

Round 2

Reviewer 1 Report

Fluoroscopy  image presented in the paper is of a low quality.

Still no imaging for percutaneous drainage